# Obesity Prevalence and Associated Socio-Demographic Characteristics and Health Behaviors in Russia and Norway

**DOI:** 10.3390/ijerph19159428

**Published:** 2022-08-01

**Authors:** Kamila Kholmatova, Alexandra Krettek, David A. Leon, Sofia Malyutina, Sarah Cook, Laila A. Hopstock, Ola Løvsletten, Alexander V. Kudryavtsev

**Affiliations:** 1Department of Community Medicine, UiT The Arctic University of Norway, N-9037 Tromsø, Norway; alexandra.krettek@uit.no (A.K.); david.leon@lshtm.ac.uk (D.A.L.); laila.hopstock@uit.no (L.A.H.); ola.lovsletten@uit.no (O.L.); alexander.v.kudryavtsev@uit.no (A.V.K.); 2International Research Competence Centre, Northern State Medical University, Troitsky Av., 51, 163069 Arkhangelsk, Russia; 3Department of Public Health, School of Health Sciences, University of Skövde, 541 28 Skövde, Sweden; 4Department of Internal Medicine and Clinical Nutrition, Institute of Medicine, Sahlgrenska Academy at University of Gothenburg, 405 30 Gothenburg, Sweden; 5Faculty of Epidemiology and Population Health, London School of Hygiene & Tropical Medicine, London WC1E 7HT, UK; sarah.cook@lshtm.ac.uk; 6Research Institute of Internal and Preventive Medicine, Branch of Institute of Cytology and Genetics, Siberian Branch of the Russian Academy of Sciences, Academician M.A. Lavrentiev Av., 17, 630090 Novosibirsk, Russia; smalyutina@hotmail.com; 7Department of Therapy, Hematology and Transfusiology, Novosibirsk State Medical University, Krasny Av., 52, 630090 Novosibirsk, Russia; 8Faculty of Medicine, National Heart and Lung Institute, Imperial College London, London SW3 6LY, UK

**Keywords:** obesity, waist-to-hip ratio, cross-sectional study, socio-demographic factors, smoking, alcohol, sex, Russia, Norway

## Abstract

Associations between obesity and socio-demographic and behavioral characteristics vary between populations. Exploring such differences should throw light on factors related to obesity. We examined associations between general obesity (GO, defined by body mass index) and abdominal obesity (AO, defined by waist-to-hip ratio) and sex, age, socio-economic characteristics (education, financial situation, marital status), smoking and alcohol consumption in women and men aged 40–69 years from the Know Your Heart study (KYH, Russia, *N* = 4121, 2015–2018) and the seventh Tromsø Study (Tromsø7, Norway, *N* = 17,646, 2015–2016). Age-standardized prevalence of GO and AO was higher in KYH compared to Tromsø7 women (36.7 vs. 22.0% and 44.2 vs. 18.4%, respectively) and similar among men (26.0 vs. 25.7% and 74.8 vs. 72.2%, respectively). The positive association of age with GO and AO was stronger in KYH vs. Tromsø7 women and for AO it was stronger in men in Tromsø7 vs. KYH. Associations between GO and socio-economic characteristics were similar in KYH and Tromsø7, except for a stronger association with living with spouse/partner in KYH men. Smoking had a positive association with AO in men in Tromsø7 and in women in both studies. Frequent drinking was negatively associated with GO and AO in Tromsø7 participants and positively associated with GO in KYH men. We found similar obesity prevalence in Russian and Norwegian men but higher obesity prevalence in Russian compared to Norwegian women. Other results suggest that the stronger association of obesity with age in Russian women is the major driver of the higher obesity prevalence among them compared to women in Norway.

## 1. Introduction

Between 1975 and 2014, the worldwide age-standardized prevalence of obesity has been estimated to have increased from 3.2% to 10.8% in men and from 6.4% to 14.9% in women [1]. In 2016, more than 13% of the population, or 650 million adults, were estimated to be obese [2]. In European countries, the prevalence of obesity has risen despite the overall improvements in population health indicators, such as life expectancy at birth and adult mortality rate [3]. The prevalence varies from 19.7% in Denmark to 27.8% in the United Kingdom [3].

Obesity is a risk factor for many diseases, including diabetes, hypertension, cardiovascular diseases, chronic kidney disease, cancer and general mortality [4,5,6,7,8]. Recent research also shows that obesity is a strong risk factor for unfavorable outcomes of coronavirus infection [9].

The substantial global rise in obesity prevalence is related to societal changes, and the imbalance between energy consumption from food and its expenditure through exercise [6,10,11,12]. In this way, obesity is largely preventable [6]. Although there are non-modifiable risk factors, such as genetic predisposition [10], identification and control of modifiable risk factors allows for effective prevention [6,10,12,13]. Despite existing knowledge about etiology, the growing worldwide prevalence of obesity and its variation across countries [1,3] indicate that obesity has socio-economic and behavioral determinants.

Socio-economic position (SEP) can be defined in different ways; for example, based on education, occupation or income. In high-income countries, there are ecological associations between lower income and education levels and higher obesity prevalence [14,15,16]. However, in low-income countries, obesity is associated with higher SEP [17]. The association of obesity with gender also varies across socio-economic levels and different cultural contexts [15,17].

According to the WHO, the prevalence of obesity has grown in Russia over the last thirty years, reaching 23.1% in 2016 [3]. These changes were observed against the background of socio-economic changes in the country [18,19]. The collapse of the Union of Soviet Socialist Republics in 1991 had dramatic economic, social and public health consequences, but the situation stabilized in the mid-2000s [18]. In parallel, the annual growth in obesity prevalence ranged between 0.1–0.2% in 1998–2004 and 0.2–0.3% in 2005–2016 [3].

Low SEP is associated with higher levels of smoking and alcohol consumption [18]. While smoking is associated with reduced levels of obesity [20,21,22], the role of alcohol consumption in weight gain is controversial and depends on the drinking pattern [23]. Therefore, these lifestyle factors can have diverse confounding effects on the associations between SEP and obesity [19].

Several epidemiologic studies have addressed obesity prevalence in Russia [18,24,25,26,27,28,29], but only a few have looked at the role of socio-demographic factors [27,28,29]. A large population-based study, “Epidemiology of cardiovascular diseases and risk factors in the regions of the Russian Federation” (ESSE-RF), described a positive association of obesity with high education and low income in men. In contrast, obesity levels were lowest in women with high education and low income [28].

One way to throw new light on the etiology of obesity is compare obesity levels and related factors in different populations. In this respect, Norway is suitable for comparison with Russia, as it is a neighboring country with similar geographic location and climate conditions but with substantially different socio-demographic characteristics. Interestingly, the prevalence of obesity among adults was the same in Norway and Russia in 2016 (23.1%) and rose in both countries in the preceding years, from 16.0% in Norway and from 19.0% in Russia in 2000 [3,30]. Income as an SEP indicator also increased in both countries during the same period. The inflation-adjusted gross domestic product (GDP) per capita increased between 2000–2019 from 14,600 to 27,000 USD in Russia and from 57,300 to 65,000 USD in Norway [31]. However, GDP in Norway was initially higher and has been more stable over time [31].

The aim of our study was to compare the prevalence of obesity and its possible associations with socio-demographic characteristics and health behaviors in Russia and Norway.

## 2. Materials and Methods

### 2.1. Study Design and Participants

This paper is based on data from two population-based studies, the Know Your Heart study in Russia and the seventh survey of the Tromsø Study in Norway.

#### 2.1.1. Know Your Heart (KYH)

During 2015–2018, the KYH cross-sectional study was conducted with random population samples of men and women aged 35–69 years from Arkhangelsk in northwestern Russia and Novosibirsk in Siberia (*N* = 5089). Further details on the KYH study design have been published previously [32]. Briefly, the participants were recruited using address databases from regional health insurance funds. Trained interviewers visited randomly selected addresses to recruit persons of the required age and sex to participate in the study. Those who agreed were interviewed at home to collect data on demographic, socio-economic and lifestyle characteristics. At the end of the interview, participants were invited to a health check at a polyclinic. The health check included a medical interview, anthropometry and other laboratory and instrumental examinations. The health check attendance was 66% for Arkhangelsk and 34% for Novosibirsk, taking as the denominator all those who were initially approached and contacted in their homes. A total of 4121 participants aged 40–69 years underwent the health check (2129 from Arkhangelsk and 1992 from Novosibirsk) and were included in this study.

#### 2.1.2. The Seventh Tromsø Study (Tromsø7)

The Tromsø Study was initiated in 1974, with repeated surveys in Tromsø municipality, Norway [33]. In 2015–2016, the seventh survey of the Tromsø Study was carried out [34]. All citizens of Tromsø aged 40 years and above (32,591) were invited by mail. Participants (21,083, 65% attendance) completed self-administered questionnaires with questions about lifestyle, socio-demographic parameters and medical anamnesis. They also underwent anthropometric, instrumental and functional measurements and provided biological samples. Further details on the study design have been published elsewhere [34]. In this study, we restricted analyses to participants aged 40–69 years (*N* = 17,646).

### 2.2. Measurements of Obesity

We defined obesity in two ways in this study: general obesity (GO) and abdominal obesity (AO).

We defined GO according to the classification from the World Health Organization (1997) as body mass index (BMI) ≥ 30.0 kg/m^2^ [35]. BMI was calculated as weight in kilograms divided by squared height in meters. In the description of GO patterns in the two countries participants’, BMI was divided into: underweight (BMI < 18.5 kg/m^2^), normal weight (BMI 18.5–24.9 kg/m^2^), overweight (25.0–29.9 kg/m^2^), obesity class I (BMI 30.0–34.9 kg/m^2^), obesity class II (BMI 35.0–39.9 kg/m^2^) and obesity class III (BMI ≥ 40.0 kg/m^2^) [35].

Abdominal or central obesity was defined as waist-to-hip ratio (WHR) > 0.9 for men and >0.85 for women [36]. WHR was calculated as the mean of two waist circumference (WC) measurements divided by the mean of two measurements of hip circumference (HC).

In both studies, weight and height were measured without shoes in light clothing by trained research technicians [32]. Weight was measured to the nearest 100 g using a TANITA BC 418 body composition analyzer (TANITA, Europe GmbH) in KYH and using an electronic digital scale (DS-B02, Dongsahn JENIX Co. Ltd., Seoul, Korea) in Tromsø7. Height was measured to the nearest millimeter using a Seca^®^ 217 portable stadiometer (Seca limited) in KYH and an electronic stadiometer (DS-103, Dongsahn JENIX Co. Ltd.) in Tromsø7. WC was measured using centimeter tape: in KYH this was undertaken at the narrowest part of the trunk to the nearest millimeter; in Tromsø7, at the level of the umbilicus [37]. A conversion equation proposed by Mason and Katzmarzyk [38] for WC in Tromsø7 (for men: narrowest = −1.19141 + 0.09503 (age) + 0.94491 (umbilicus); for women: narrowest = −1.02517 + 0.03207 (age) + 0.90184 (umbilicus)) was used to compare WC in the two studies [37]. HC was measured at the widest part of the hips in both studies.

### 2.3. Socio-Demographic Characteristics

In KYH, the following socio-demographic variables were collected: age (years); sex (male/female); marital status (living together with a spouse/partner in a registered marriage, living together with spouse/partner but not in a registered marriage, divorced or separated, widower, never married); education (incomplete secondary, complete secondary, professional no secondary, professional and secondary, specialized secondary (college, 3 years), incomplete higher, higher); and self-perceived financial situation of the household measured on a categorical scale with six options (not even enough money for food, it is difficult to make ends meet; enough money for food, but difficult to afford clothes and other items; enough money for food and clothes, but difficult to buy large domestic appliances; can afford to buy large domestic appliances, but difficult to buy a large new car; can afford to buy a large new car, but difficult to buy a flat or a house; no financial constraints, can afford to buy a flat or a house).

In Tromsø7, the corresponding socio-demographic variables were: age (years), sex (male/female), living with a spouse/partner (yes/no); education (primary/partly secondary education (up to 10 years of schooling); upper secondary education (a minimum of 3 years); tertiary education, short (college/university less than 4 years); tertiary education, long (college/university 4 years or more)); and annual total income of the household (less than 150,000 NOK, 150,000–250,000 NOK, 251,000–350,000 NOK, 351,000–450,000 NOK, 451,000–550,000 NOK, 551,000–750,000 NOK, 751,000–1,000,000 NOK, more than 1,000,000 NOK).

For the comparisons of KYH with Tromsø7, the socio-demographic variables were harmonized by recoding them into a unified format: age (5 year age groups); marital status: living with spouse/partner (yes/no); education: primary/secondary education (incomplete secondary, complete secondary, professional no secondary and professional and secondary in KYH; primary/partly secondary education and upper secondary education in Tromsø7) and college/university education (specialized secondary, incomplete higher, and higher in KYH; short and long tertiary education in Tromsø7); and financial situation of the household: level 1 (not enough/enough money for food but difficult to afford clothes in KYH; ≤350,000 NOK in Tromsø7), level 2 (enough money for food and clothes but difficult to buy large domestic appliances in KYH; 351,000–1,000,000 NOK in Tromsø7) and level 3 (enough money for large domestic appliances and higher categories in KYH; >1,000,000 NOK in Tromsø7).

### 2.4. Health Behaviours

In both studies, questions on health behaviors included daily tobacco/cigarette smoking (never, ex-smoker, current smoker). Both studies used the Alcohol Use Disorders Identification Test (AUDIT) [39], from which we took data on frequency of alcohol use, categorized as “2 or more times per week” or less often, and data on the number of standard alcohol units normally taken per drinking occasion, categorized as “5 or more” (binge drinking) or less.

### 2.5. Statistical Analysis

Data were expressed as mean values and standard deviations (SD) for continuous variables. Absolute numbers (Abs) and proportions (%) were reported for categorical variables. Age-standardized prevalence of GO and AO was calculated based on the European Standard Population 2013 with 5 year age intervals. Within-study assessments of the associations of GO and AO with socio-demographic and lifestyle characteristics were performed in the KYH and Tromsø7 datasets, separately. Logistic regressions with three-step entry of covariates were used. At step one (model 1), only adjustments for age were made. At the second step (model 2), we added SEP characteristics. At the third step (model 3), we entered all analyzed SEP and health behavior variables. In these and further analyses, we excluded all participants with missing values for any of the covariates, so the studied samples comprised 4024 participants from KYH and 15,892 from Tromsø7. Between-study comparisons of the strength of associations of GO and AO with covariates of interest were performed in Models 3 repeated with pooled KYH and Tromsø7 data and added “study” variable. These comparisons were made by assessing interactions of the “study” variable with all other entered covariates. Interactions were assessed by comparing models with and without interaction terms using likelihood ratio tests. We used similar regressions (models 1–3) for the KYH and Tromsø7 pooled dataset to assess the effects of the socio-demographic and health behavior characteristics on the between-study differences in GO and AO. Statistical analysis was performed using STATA V.16 (StataCorp, College Station, TX, USA).

## 3. Results

### 3.1. Socio-Demographic Characteristics and Health Behaviors

The average age of men and women in KYH (55.7 and 55.4 years) was slightly higher than that of men and women in Tromsø7 (53.8 and 53.6 years) (Table 1). There were more men and women in KYH with college or university education (70.0% and 78.3%) compared to Tromsø7 (50.5% and 55.7%), but higher proportions of KYH participants belonged to the low financial level category (17.8% vs. 6.5% for men and 22.7% vs. 10.8% for women). Men living with their spouse/partner was marginally more common in KYH compared to Tromsø7 (84.8% vs. 81.8%), but among women it was much more common in Tromsø7 compared to KYH (75.4% vs. 56.9%). The proportion of current smokers was higher among KYH men than Tromsø7 men (35.7% vs. 13.8%), but similar among women (15.0% vs. 15.3%). Drinking alcohol ≥ 2 times per week was more common in Tromsø7 compared to KYH for men (34.2% vs. 20.7%) and women (27.3% vs. 2.81%), while drinking ≥5 drinks per occasion was more common in KYH (35.7% vs. 14.8% among men and 8.2% vs. 4.2% among women).

### 3.2. Prevalence of Obesity

The age-standardized prevalence of GO (BMI ≥ 30 kg/m^2^) was higher among KYH women (36.7%) compared to women in Tromsø7 (22.0%), but it was very similar among men (26.0% and 25.7%, respectively). In KYH women, GO prevalence increased steeply with age. There was no evidence of equivalent increases with age in Tromsø7 women, nor in men in either study (Figure 1 and Appendix A).

The age-standardized prevalence of overweight was 44.2% in KYH and 50.9% in Tromsø7 men and 31.8% and 36.3% in KYH and Tromsø7 women. The age-standardized prevalence of normal weight was 28.1% and 23.3% in men and 30.3% and 40.8% in women in KYH and Tromsø7, respectively. The proportion of KYH women who were overweight was 26.3% at 40–44 years and 37.6% at 65–69 years (Figure 2a and Appendix A), while in Tromsø7 women, the proportions were 33.5% and 40.5%, respectively. Normal weight was found in 44.4% of Tromsø7 women at 40–44 years and in 35.9% at 65–69 years, while in KYH women the corresponding proportions were 47.7% and 13.8%. In both studies, the proportions of men who were overweight and had normal weight were stable across age groups (Figure 2b and Appendix A).

The age-standardized prevalence of AO (WHR > 0.9 for men and >0.85 for women) was higher in KYH compared to Tromsø7 women (44.2% vs. 18.4%) and similar (74.8% vs. 72.2%) in KYH and Tromsø7 men (Figure 3 and Appendix A). Unlike for GO, the prevalence increased with age in both sexes and studies. Among participants without GO, 67.8% vs. 63.9% of men and 30.9% vs. 12.2% of women had AO in KYH vs. Tromsø7, respectively (Appendix A). The majority of those with AO but without GO were overweight (76.5% men and 79.7% women in KYH, 81.2% men and 75.6% women in Tromsø7) (Appendix A).

### 3.3. Age of Participants

In Tromsø7 men, but not in KYH men, odds ratios (ORs) of GO decreased with age in the unadjusted model (model 1) and with adjustments for SEP covariates (model 2) and health behaviors (model 3, *p*-trend = 0.049) (Table 2a). However, there were no significant between-study differences in GO trends by age in the fully adjusted models (*p* = 0.42). Odds of AO were higher with older age for men in both studies before and after all adjustments (Table 3a), but the trend was steeper for Tromsø7 men (*p* interaction < 0.001). For KYH women (Table 2b), ORs of GO went up with age in unadjusted and adjusted models. For Tromsø7 women, a downward trend in ORs of GO with age was observed after adjustments for SEP and lifestyle covariates (*p*-trend = 0.026). Among women, the between-study differences in trends of GO with age were highly significant. Odds of AO were higher with higher age for women in both studies, irrespectively of adjustments (Table 3b). The trend was steeper among KYH women (*p* < 0.001).

### 3.4. Education

In Tromsø7, men with primary and secondary education had increased odds of GO compared to men with higher education in all models (Table 2a). Similarly, Tromsø7 men with lower education had higher odds of AO (Table 3a). Education did not show significant associations with odds of GO and AO in KYH men. However, there were no between-study differences in the strength of the association of education with GO and AO. In both KYH and Tromsø7, women with lower education had higher ORs of GO and AO after adjustments for age and other covariates (Table 2b and Table 3b). Associations of education with GO and AO did not differ between studies, neither in men nor in women.

### 3.5. Marital Status

Men in KYH, but not in Tromsø7, had higher odds of GO if living with spouse/partner regardless of the adjustments (Table 2a). The association was the opposite in Tromsø7 men but only before controlling for other SEP covariates. Similarly, living with spouse/partner was negatively associated with AO in Tromsø7 women but only before adjustments for other SEP covariates (Table 3b). Living with spouse/partner had a stronger association with GO among KYH men compared to men in Tromsø7 (*p* = 0.005) but not among women and not with AO among both sexes.

### 3.6. Financial Situation

Among Tromsø7 men, but not KYH men (Table 2a), there were higher odds of GO at the middle and lower financial levels compared to the upper level in the age-adjusted model, which were gradually attenuated to non-significance by adjustments for other SEP covariates and health behaviors. Middle-income Tromsø7 men also had elevated odds of AO, which were attenuated by the same adjustments but sustained statistical significance. Women at the lower financial level had higher odds of GO and AO in both KYH and Tromsø7 compared to those at the upper financial level, irrespectively of adjustments (Table 2b and Table 3b). Significantly elevated odds of GO and AO were also found in Tromsø7 women at the middle vs. higher financial level, and these persisted through all the adjustments. There were no between-study differences in the strength of the association of financial situation with GO and AO.

### 3.7. Smoking

In the fully adjusted models for men, both KYH and Tromsø7 current smokers had lower odds of GO compared to those who had never smoked (Table 2a). Smoking men in Tromsø7, but not in KYH, had elevated odds of AO (Table 3a), giving a difference in the association strength (*p* = 0.012). Among women, lower odds of GO for current smokers were observed in Tromsø7 only (Table 2b), while odds of AO were similarly elevated in smokers relative to those who had never smoked in both studies. Ex-smokers among men and women in Tromsø7 had higher odds of GO (Table 2a,b), while elevated odds of AO were observed in ex-smoking KYH men and Tromsø7 men and women (Table 3a,b). The strength of associations between smoking and AO did not differ between the studies.

### 3.8. Alcohol Consumption

Frequent male drinkers had increased odds of GO in KYH and reduced odds of GO in Tromsø7 (Table 2a), reflecting the opposite associations (*p* < 0.001). A similar situation with respect to GO was observed in KYH vs. Tromsø7 women (*p* = 0.001), as well as relative to AO in KYH vs. Tromsø7 men (*p* = 0.007) (Table 2b and Table 3a). In both studies, binge drinkers, both male and female, had elevated odds of GO and AO, although the OR of GO in KYH women and the OR of AO in Tromsø7 women did not reach statistical significance (Table 2a,b and Table 3a,b). The association of binge drinking with AO in Tromsø7 men was stronger compared to KYH men (*p* = 0.015).

### 3.9. Country Effect on Associations of Obesity with Socio-Demographic and Lifestyle Characteristics

Men in KYH and Tromsø7 showed no significant differences in odds of GO and AO in age-adjusted models, and this did not change after adjustments for SEP characteristics and health behaviors (Table 4). Women in KYH had substantially increased odds of GO (OR = 2.20) and AO (OR = 3.86) compared to Tromsø7 women in age-adjusted models. The ORs were insubstantially attenuated by adjustments for SEP and behavioral covariates. 

## 4. Discussion

Compared to Norwegian women, Russian women demonstrated a higher prevalence of GO and AO, while there was no difference for men. Older age was stronger associated with higher odds of GO and AO in Russian vs. Norwegian women and with higher odds of AO in Norwegian vs. Russian men. We observed a stronger association between GO and living with a spouse or partner in Russian vs. Norwegian men. Furthermore, Norwegian men showed a positive association of current smoking with AO vs. no association in Russian men. Drinking alcohol two or more times per week had a positive vs. negative association with GO and AO in Russian vs. Norwegian men and with GO in Russian vs. Norwegian women. Binge drinking was more strongly associated with AO in Norwegian vs. Russian men.

Higher obesity prevalence at an older age has been described previously [14,26,36,40,41,42,43,44,45]. It may be explained by metabolic changes at the age of 40–69 years in both sexes [46] and by menopause-related hormonal changes in women [36,47,48]. The higher GO prevalence we observed in Russian women between 40 and 69 years is consistent with findings from other Russian studies [25,26,28,49]. Possible explanations of the higher obesity prevalence in Russian versus Norwegian women may be grounded in a more pronounced traditional female gender role [50] and related daily life contexts of ageing women in Russia [47,51].

The lower levels of GO found with higher age in Norwegian men and women might indicate a higher obesity prevalence among younger age groups (<40 years of age) in this population [52,53,54]. A possible explanation could be a cohort effect [54,55,56,57], but it was not possible to assess this objectively in our study.

We saw a higher GO prevalence in Tromsø7 men compared to women, while KYH women had higher GO prevalence compared to men. Sex differences, such as that found in Tromsø7, were previously found in high-income European studies [40,41,42,58,59], while the higher GO prevalence in women compared to men was previously described in low- and middle-income countries [60,61] and in previous studies in Russia [26,28,49,62,63,64]. This supports our abovementioned presumption that socio-economic and cultural factors contribute to the higher obesity levels in Russian vs. Norwegian women, but they make no difference when comparing Russian men to Norwegians.

Obesity can be defined using BMI only. However, this approach does not consider the distribution of adipose tissue in the body and may not detect AO [48,65,66]. This is in line with our finding concerning the higher AO prevalence compared to GO prevalence among men in both studies and is consistent with prior research comparing different obesity measurements [40,53,65]. Our data have also shown that up to 60% of men and 30% of women without GO could be misclassified as non-obese if we did not consider AO. In addition, there is evidence that obesity-related disorders can be more prevalent in non-obese people with high visceral fat accumulation compared to those with GO [67]. This indicates the importance of assessing both obesity types.

Trying to apprehend and understand our findings concerning the higher AO compared to GO, we also discovered that the majority (80%) of the total participants in our study with AO but without GO were overweight according to the BMI-based classification [35]. This partly explains our other finding concerning the overall higher AO prevalence in men (73%) compared to women (24%), as we could see that a larger proportion of men were overweight (50%) relative to women (36%).

Another important observation made by prior researchers is that AO prevalence depends on how it is measured. It can be higher in women than in men if assessed using WC data [27,53,54], but it can be the opposite, as in our study, when the assessments are based on WHR or waist-to-height ratio (WHeR) [40,68].

Assessing AO is important because it is commonly combined with other components of metabolic syndrome [48,66] and is a stronger predictor of CVD and diabetes compared to GO [24,28,69]. WHR showed the strongest associations with overall and cardiovascular disease mortality compared to other AO indicators (WC or WHeR) [27,48,65,66]. Obesity-related complications are less common in patients with a predominant deposition of fat in the buttocks and thighs; thus, increased HC is protective in both genders when controlling for WC and should be included in obesity anthropometric measurements [24,27,65].

Previous studies have shown that, in high-income countries, women with higher income had lower levels of obesity [40,43,44]. Lower levels of education were associated with AO either in both sexes [40] or in men only [44]. These associations could be explained by better knowledge of healthy nutrition and obesity-related health risks among people with higher education and, commonly associated, higher income [40,44,62]. Higher income may be associated with higher availability for the components of a healthy diet and lower barriers to having a healthier diet [15,17,63], as well as being more physically active [15,70].

In our study, no university education, low income and middle income were associated with higher odds of GO and AO in men and women in Norway, but in Russia the association of increased odds of GO and AO with low income and education was only observed in women. However, the differences in the strength of the association between education and financial situation and obesity were not statistically confirmed for either of the sexes. This is generally in line with prior findings of SEP associations with obesity [40,44,62,63], but the results of our international comparisons may reflect the fact that the differences in obesity-related knowledge and behaviors are not as distinguished between Russian men with different SEPs compared to those of Norwegian men with different SEP levels. Interestingly, our comparisons of the association between obesity and education and income in the two countries could have been affected by the overall difference in income levels between Russia and Norway [31]. For example, being middle-income in Norway is associated with an objectively different standard of living compared to being middle-income in Russia. Furthermore, our comparisons of SEP–obesity associations could have been affected by the ways in which the compared categories of education and income were derived from the different original variables in the two studies.

It has previously been described that the association between SEP and obesity is different in low- and high-income countries and becomes inverted when countries transit into a higher income category [15,16,70,71]. In 2020, LM Jaacks et al. proposed the theory of obesity transition, comprising four stages based on a given country’s socio-demographic characteristics and obesity patterns [16]. Based on this theory, Norway is at the third stage of the obesity transition, as there is no difference in GO prevalence between men and women and high SEP is associated with reduced GO levels in both men and women. Perhaps Norway is even approaching the end of the third stage, given that the increase in obesity prevalence is slowing down [3]. In contrast, Russia is at the second stage, as GO prevalence among women is higher compared to men and SEP shows little relationship with GO in men.

In our study, living with a spouse or partner was associated with GO only in Russian men. Several prior studies report that both sexes showed higher obesity prevalence when married or living with partner [40,72], while others did not find any association [43]. Some results were consistent with ours in showing that being married was associated with obesity in men only [41,62]. Possible explanations could be that Russian married men eat more regularly, have a larger number of meals and consume more sugar- or salt-rich homemade preparations (traditional dishes in Russia) [63,73]. Conversely, single men may eat less regularly and are more commonly smokers [73]. The observed between-country difference in the strength of the association between GO and marital status in men may be due to Russian–Norwegian differences in the gender roles of married men and women [74], but this hypothesis requires further investigation.

Some studies have suggested that current smokers have lower BMI compared to those who have never smoked and ex-smokers [20,21,22], whereas others have found no associations of this kind [75]. Conversely, current smoking has a positive association with AO [20,75]. The highest risk of AO has been found for former smokers and is explained by weight gain after smoking cessation [21]. There are also contradictory data on the association of GO with alcohol intake. This may be due to the varying types of beverages consumed and different alcohol-associated food intake patterns [23,76,77,78,79]. For example, drinking wine may have a protective effect on weight gain compared to consumption of spirits, which can lead to overweight and obesity [76]. Several studies show higher obesity levels with higher volumes of alcohol consumed [78,79,80,81,82], while others found that low but regular intake of alcohol protected against weight gain [76,77,83]. However, binge drinking has been described as positively associated with GO and AO [82], predominantly in men [80,81].

Being concerned about the possible mediating effect of health behaviors in the associations between socio-demographic characteristics and obesity, we included smoking and alcohol consumption characteristics in our analyses. Entering these variables in regression models resulted in only modest attenuations of the ORs of GO and AO for education and financial situation among men and women in Tromsø7 but not in KYH. This may have reflected between-country differences in the associations between SEP and health behaviors and partially confirms our earlier hypothesis that obesity-related health behaviors may not be as variable between Russian men with different SEPs compared to the variation in Norwegians with SEP differences.

The differences in associations of obesity with smoking and alcohol discovered reflect the varying lifestyle patterns of the two populations. For instance, we found smoking to be more prevalent among Russian men and, thus, it may be “less selective” in its health effects. Specifically, it may have weaker associations with other unhealthy behaviors compared to among Norwegian men. An earlier study comparing alcohol consumption in the KYH and Tromsø7 study populations showed that hazardous and problem drinking were more prevalent in KYH men compared to men in Tromsø7, but it was the opposite in women [84]. This may indicate more complex between-study differences in the alcohol–obesity associations compared to what we could detect with a relatively rough categorization of participants by alcohol-related quantity and frequency variables.

The low prevalence (2.81%) of drinking alcohol ≥2 times per week among KYH women could be connected with underreporting of drinking frequency. Although underreporting of alcohol consumption may be inherent for both samples, previous studies have shown that Russian women underreport alcohol consumption compared to their Norwegian and Finnish counterparts [85,86].

### Strengths and Limitations

We examined the prevalence of both GO and AO in Russian and Norwegian adult populations and described socio-demographic and lifestyle correlates of both conditions comparatively, thus shedding new light on risk groups and preventive measures to be prioritized. One strength of the study is that we used data from two large population-based studies conducted during the same time period. Another strength is that anthropometric measurements were undertaken by trained personnel and not self-reported, which is prone to bias [87].

An important limitation is that we could only take into account a limited spectrum of health behaviors. Smoking and alcohol consumption were the only obesity-related lifestyle characteristics that were measured comparably in the two studies. For this reason, we did not study the effects of diet and physical activity, although the imbalance between high energy intake from food and low levels of physical activity is a major etiological factor for obesity development. 

Another limitation is that harmonization of the variables used in KYH and Tromsø7 for between-study comparisons included some approximations and assumptions. For instance, WC was measured using different anatomical landmarks, so the comparisons might have been partially biased, although we used a previously proposed and validated conversion equation. The harmonized education variable was derived from categorical variables reflecting different education systems in the two countries. Questions about the financial well-being of families implied self-reported assessments and were also asked in different ways. In Norway, respondents indicated the annual household income, while in Russia participants were asked to assess their capacities to buy certain goods. Therefore, the studied associations of these variables with obesity could have been underestimated.

The KYH study was based on data from two cities in Russia (Arkhangelsk in the European north and Novosibirsk in the south of Western Siberia), while Tromsø7 was based on data from Tromsø municipality in the north of Norway. Tromsø municipality is demographically and economically similar to the rest of Norway [88,89]. The populations of Arkhangelsk and Novosibirsk were described as comparable in age and education distribution to the total Russian urban population [32]. Therefore, the sampled populations should not have major socio-demographic and lifestyle differences compared to the entire Russian and Norwegian populations.

Finally, the attendance was 65% in Tromsø7, 66% in the Arkhangelsk part of the KYH study and 34% in its Novosibirsk part. Therefore, our study populations might not have been fully representative of the underlying target populations due to potential non-response bias.

## 5. Conclusions

The prevalence of GO and AO was higher in Russian compared to Norwegian women but did not differ in men. Older age was more strongly associated with GO and AO in Russian compared to Norwegian women and with AO in Norwegian relative to Russian men. Low SEP increased the odds of both obesity types among men and women in Norway and among women in Russia, but not in Russian men. However, living with a spouse or partner was associated with GO in Russian but not Norwegian men. Current smoking was associated with reduced odds of GO in Russian men and increased odds of AO among men in Norway. Frequent alcohol drinking was associated with increased odds of GO in Russian men but reduced odds of GO in Norwegian men, and the same discrepancy in drinking frequency associations with GO was observed in Russian compared to Norwegian women. Conversely, drinking large volumes per occasion was more strongly associated with AO in Norwegian vs. Russian men. These differences shed new light on the socio-demographic and lifestyle predictors of obesity to be addressed in each of the two countries.

## Figures and Tables

**Figure 1 ijerph-19-09428-f001:**
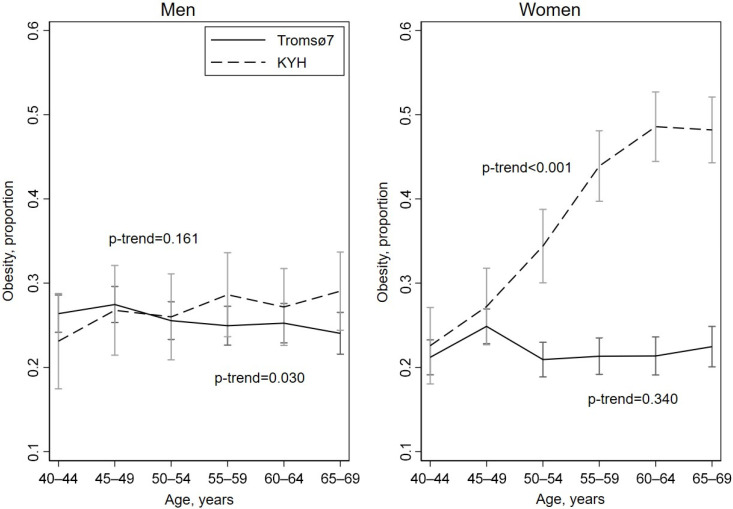
Prevalence of general obesity (BMI ≥ 30 kg/m^2^) with 95% confidence intervals by study, sex and age.

**Figure 2 ijerph-19-09428-f002:**
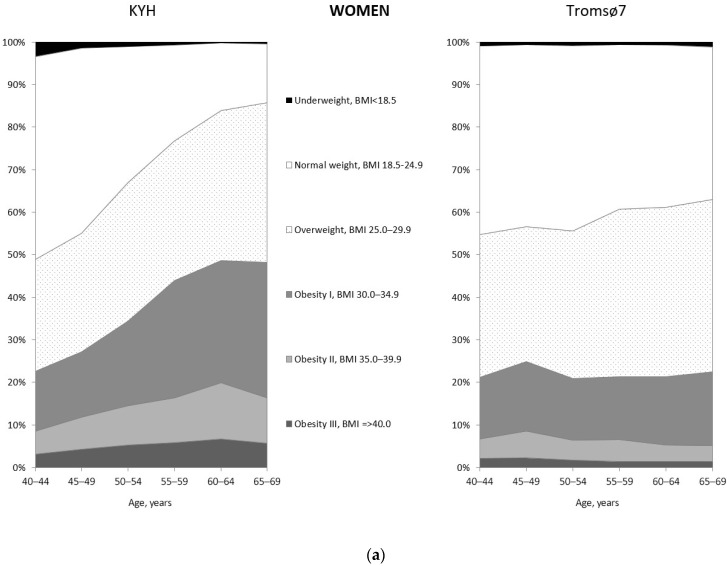
Distribution of women (**a**) and men (**b**) in the compared populations by study, age and BMI categories.

**Figure 3 ijerph-19-09428-f003:**
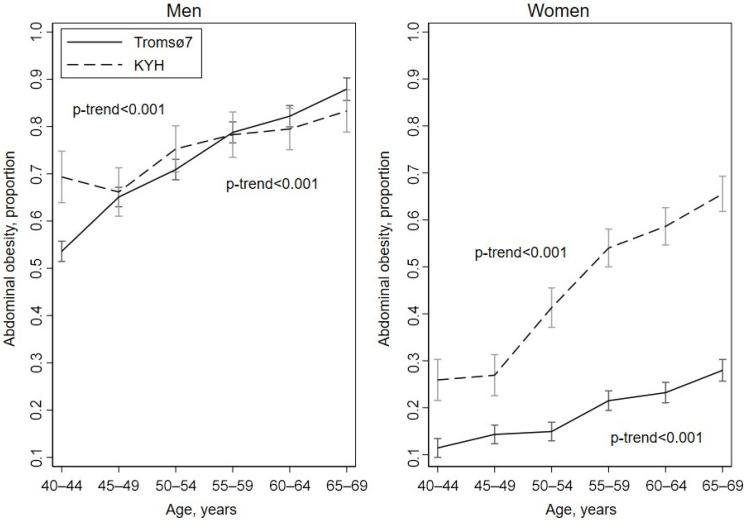
Prevalence of abdominal obesity (WHR > 0.9 for men and >0.85 for women) with 95% confidence intervals by study, sex and age.

**Table 1 ijerph-19-09428-t001:** Socio-demographic characteristics and health behaviors of studied populations by study and sex.

	Men		Women	
	KYH † (*N* = 1732)	Tromsø7 ‡ (*N* = 8346)	KYH † (*N* = 2389)	Tromsø7 ‡ (*N* = 9300)
Age, years (Mean, SD)	55.7 (8.5)	53.8 (8.5)	55.4 (8.7)	53.6 (8.4)
	Abs (%)		Abs (%)	
Age groups, years				
˗ 40–44	226 (13.1)	1473 (17.7)	351 (14.7)	1678 (18.0)
˗ 45–49	254 (14.7)	1581 (18.9)	349 (14.6)	1700 (18.3)
˗ 50–54	279 (16.1)	1434 (17.2)	381 (16.0)	1705 (18.3)
˗ 55–59	290 (16.7)	1356 (16.3)	411 (17.2)	1540 (16.6)
˗ 60–64	347 (20.0)	1320 (15.8)	423 (17.7)	1420 (15.3)
˗ 65–69	336 (19.4)	1182 (14.2)	474 (19.8)	1257 (13.5)
Education				
˗ Primary/secondary	519 (30.0)	4090 (49.5)	519 (21.7)	4081 (44.3)
˗ College/university	1213 (70.0)	4173 (50.5)	1870 (78.3)	5128 (55.7)
Living with spouse/partner (yes)	1469 (84.8)	6609 (81.8)	1360 (56.9)	6532 (75.4)
Financial situation, level				
˗ Lower	301 (17.8)	537 (6.5)	535 (22.7)	968 (10.8)
˗ Middle	1284 (76.0)	4991 (60.8)	1709 (72.6)	5769 (64.4)
˗ Upper	104 (6.2)	2684 (32.7)	110 (4.7)	2226 (24.8)
Smoking status				
˗ Never	497 (28.7)	3705 (45.3)	1723 (72.1)	3897 (42.7)
˗ Ex-smoker	617 (35.6)	3349 (40.9)	307 (12.9)	3840 (42.0)
˗ Current smoker	618 (35.7)	1133 (13.8)	358 (15.0)	1402 (15.3)
Drinking alcohol 2+ times per week (yes)	357 (20.7)	2845 (34.2)	67 (2.81)	2524 (27.3)
Drinking 5+ alcohol drinks per occasion (yes)	615 (35.7)	1225 (14.8)	195 (8.19)	387 (4.21)

Abs—absolute number of participants with corresponding characteristic. † Missing data in KYH: financial situation—78 (1.9%), drinking alcohol 2+ times per week—13 (0.3%), drinking 5+ alcohol drinks per occasion—18 (0.4%). ‡ Missing data in Tromsø7: education—77 (1.0%), living with partner—896 (5.1%), financial situation—471 (2.7%), smoking status—320 (1.8%), drinking alcohol 2+ times per week—72 (0.4%), drinking 5+ alcohol drinks per occasion—197 (1.1%).

**Table 2 ijerph-19-09428-t002:** (a). Associations of socio-demographic and behavioral characteristics with general obesity (BMI ≥ 3 0 kg/m^2^) by study among men. (b) Associations of socio-demographic and behavioral characteristics with general obesity (BMI ≥ 30 kg/m^2^) by study among women.

**(a)**								
		**KYH, OR (95% CI)**			**Tromsø7, OR (95% CI)**			** *p* ** **-** **Value for Interaction ^d^**
		**Model 1 ^a^**	**Model 2 ^b^**	**Model 3 ^c^**	**Model 1 ^a^**	**Model 2 ^b^**	**Model 3 ^c^**	
Age, years	- 40–44	1.00	1.00	1.00	1.00	1.00	1.00	-
	- 45–49	1.15 (0.75–1.76)	1.15 (0.75–1.77)	1.21 (0.79–1.87)	1.11 (0.94–1.31)	1.08 (0.91–1.28)	1.10 (0.93–1.30)	-
	- 50–54	1.14 (0.75–1.72)	1.14 (0.75–1.73)	1.22 (0.80–1.86)	0.97 (0.82–1.15)	0.92 (0.77–1.09)	0.97 (0.82–1.16)	-
	- 55–59	1.31 (0.88–1.97)	1.29 (0.86–1.95)	1.39 (0.92–2.10)	0.97 (0.81–1.15)	0.90 (0.76–1.08)	0.97 (0.81–1.16)	0.423
	- 60–64	1.19 (0.80–1.77)	1.14 (0.76–1.70)	1.19 (0.79–1.79)	0.93 (0.78–1.11)	0.87 (0.72–1.04)	0.95 (0.79–1.14)	-
	- 65–69	1.34 (0.90–1.99)	1.31 (0.88–1.96)	1.33 (0.89–2.00)	0.87 (0.73–1.05)	0.79 (0.66–0.96)	0.87 (0.71–1.05)	-
	*p*-value for trend	0.161	0.244	0.256	0.030	0.001	0.049	-
Education	- College/university	1.00	1.00	1.00	1.00	1.00	1.00	-
	- Primary/secondary	1.05 (0.83–1.33)	1.08 (0.85–1.37)	1.18 (0.92–1.50)	1.57 (1.42–1.75)	1.51 (1.35–1.68)	1.36 (1.21–1.52)	0.296
Living with spouse/partner	- No	1.00	1.00	1.00	1.00	1.00	1.00	-
	- Yes	1.63 (1.17–2.27)	1.64 (1.17–2.29)	1.58 (1.13–2.21)	0.86 (0.75–0.98)	0.90 (0.78–1.04)	0.93 (0.81–1.08)	0.005
Financial situation, level	- Upper	1.00	1.00	1.00	1.00	1.00	1.00	-
	- Middle	1.05 (0.67–1.66)	1.02 (0.64–1.62)	1.02 (0.64–1.63)	1.37 (1.23–1.54)	1.18 (1.05–1.34)	1.12 (0.99–1.27)	-
	- Lower	0.98 (0.59–1.64)	0.99 (0.59–1.66)	1.09 (0.65–1.85)	1.39 (1.09–1.75)	1.04 (0.80–1.36)	1.00 (0.77–1.31)	0.625
Smoking status	- Never	1.00		1.00	1.00		1.00	-
	- Ex-smoker	1.17 (0.90–1.52)		1.09 (0.83–1.42)	1.46 (1.30–1.63)		1.32 (1.18–1.48)	-
	- Current smoker	0.61 (0.46–0.81)		0.56 (0.42–0.74)	1.04 (0.88–1.23)		0.80 (0.68–0.96)	0.100
Drinking alcohol 2+ times per week	- No	1.00		1.00	1.00		1.00	-
	- Yes	1.48 (1.14–1.91)		1.48 (1.14–1.92)	0.61 (0.54–0.68)		0.66 (0.59–0.74)	<0.001
Drinking 5+ alcohol drinks per occasion	- No	1.00		1.00	1.00		1.00	-
	- Yes	1.25 (0.99–1.56)		1.34 (1.06–1.69)	1.84 (1.61–2.10)		1.66 (1.44–1.91)	0.118
(b)								
		**KYH, OR (95% CI)**			**Tromsø7, OR (95% CI)**			** *p* ** **-** **Value for Interaction ^d^**
		**Model 1 ^a^**	**Model 2 ^b^**	**Model 3 ^c^**	**Model 1 ^a^**	**Model 2 ^b^**	**Model 3 ^c^**	
Age, years	- 40–44	1.00	1.00	1.00	1.00	1.00	1.00	-
	- 45–49	1.24 (0.88–1.76)	1.28 (0.90–1.82)	1.30 (0.91–1.84)	1.21 (1.02–1.43)	1.17 (0.98–1.39)	1.21 (1.02–1.44)	-
	- 50–54	1.70 (1.22–2.37)	1.70 (1.22–2.38)	1.75 (1.25–2.45)	0.96 (0.80–1.14)	0.88 (0.74–1.06)	0.94 (0.79–1.13)	-
	- 55–59	2.66 (1.93–3.65)	2.54 (1.84–3.50)	2.60 (1.88–3.60)	0.94 (0.78–1.13)	0.83 (0.69–1.00)	0.91 (0.75–1.10)	<0.001
	- 60–64	3.17 (2.31–4.35)	2.96 (2.15–4.08)	3.08 (2.22–4.27)	0.99 (0.82–1.19)	0.84 (0.69–1.01)	0.93 (0.77–1.13)	-
	- 65–69	3.18 (2.33–4.34)	2.97 (2.17–4.06)	3.08 (2.23–4.25)	1.04 (0.86–1.26)	0.82 (0.67–1.00)	0.90 (0.74–1.11)	-
	*p*-value for trend	<0.001	<0.001	<0.001	0.340	<0.001	0.026	-
Education	- College/university	1.00	1.00	1.00	1.00	1.00	1.00	-
	- Primary/secondary	1.57 (1.28–1.93)	1.53 (1.25–1.88)	1.54 (1.26–1.90)	1.59 (1.43–1.78)	1.44 (1.28–1.62)	1.37 (1.21–1.54)	0.317
Living with spouse/partner	- No	1.00	1.00	1.00	1.00	1.00	1.00	-
	- Yes	0.94 (0.79–1.12)	1.00 (0.84–1.20)	1.00 (0.84–1.19)	0.89 (0.79–1.01)	0.99 (0.87–1.14)	1.00 (0.87–1.14)	0.996
Financial situation, level	- Upper	1.00	1.00	1.00	1.00	1.00	1.00	-
	- Middle	1.25 (0.81–1.93)	1.24 (0.80–1.91)	1.25 (0.81–1.94)	1.67 (1.46–1.91)	1.49 (1.29–1.72)	1.40 (1.21–1.62)	-
	- Lower	1.73 (1.10–2.74)	1.66 (1.04–2.64)	1.68 (1.06–2.68)	1.84 (1.50–2.26)	1.50 (1.18–1.91)	1.37 (1.07–1.75)	0.081
Smoking status	- Never	1.00		1.00	1.00		1.00	-
	- Ex-smoker	1.19 (0.91–1.54)		1.14 (0.87–1.48)	1.27 (1.13–1.43)		1.21 (1.08–1.37)	-
	- Current smoker	1.02 (0.79–1.31)		0.89 (0.68–1.15)	0.95 (0.81–1.13)		0.76 (0.64–0.91)	0.500
Drinking alcohol 2+ times per week	- No	1.00		1.00	1.00		1.00	-
	- Yes	1.53 (0.92–2.54)		1.58 (0.95–2.63)	0.51 (0.45–0.58)		0.55 (0.48–0.63)	0.001
Drinking 5+ alcohol drinks per occasion	- No	1.00		1.00	1.00		1.00	-
	- Yes	1.32 (0.97–1.81)		1.26 (0.91–1.75)	1.84 (1.45–2.33)		1.68 (1.31–2.14)	0.173

^a^ Model 1—age-adjusted for all variables except age; ^b^ model 2—adjusted for age and socio-economic variables; ^c^ model 3—adjusted for age, socio-economic variables and health behaviors; ^d^ likelihood ratio test for interaction with “study” variable in model 3 repeated with pooled KYH and Tromsø7 data and the introduced “study” variable.

**Table 3 ijerph-19-09428-t003:** (a) Associations of socio-demographic and behavioral characteristics with abdominal obesity (WHR > 0.9) by study among men. (b) Associations of socio-demographic and behavioral characteristics with abdominal obesity (WHR > 0.85) by study among women.

(a)								
		**KYH, OR (95% CI)**			**Tromsø7, OR (95% CI)**			***p*-Value for Interaction ^d^**
		**Model 1 ^a^**	**Model 2 ^b^**	**Model 3 ^c^**	**Model 1 ^a^**	**Model 2 ^b^**	**Model 3 ^c^**	
Age, years	- 40–44	1.00	1.00	1.00	1.00	1.00	1.00	-
	- 45–49	0.82 (0.55–1.21)	0.82 (0.55–1.22)	0.82 (0.55–1.23)	1.60 (1.37–1.86)	1.56 (1.34–1.82)	1.58 (1.36–1.85)	-
	- 50–54	1.36 (0.91–2.04)	1.37 (0.92–2.05)	1.40 (0.93–2.10)	2.06 (1.75–2.41)	1.96 (1.67–2.30)	2.04 (1.74–2.41)	-
	- 55–59	1.54 (1.03–2.31)	1.54 (1.03–2.32)	1.60 (1.06–2.41)	3.22 (2.71–3.83)	3.04 (2.56–3.62)	3.22 (2.69–3.84)	<0.001
	- 60–64	1.66 (1.12–2.46)	1.64 (1.11–2.45)	1.66 (1.11–2.49)	3.97 (3.31–4.75)	3.74 (3.12–4.49)	3.90 (3.23–4.69)	-
	- 65–69	2.24 (1.48–3.39)	2.26 (1.49–3.43)	2.26 (1.48–3.46)	6.28 (5.08–7.77)	5.77 (4.65–7.16)	6.13 (4.92–7.63)	-
	*p*-value for trend	<0.001	<0.001	<0.001	<0.001	<0.001	<0.001	-
Education	- College/university	1.00	1.00	1.00	1.00	1.00	1.00	-
	- Primary/secondary	0.99 (0.77–1.27)	1.01 (0.78–1.30)	1.05 (0.81–1.37)	1.72 (1.55–1.91)	1.63 (1.46–1.82)	1.38 (1.23–1.55)	0.062
Living with spouse/partner	- No	1.00	1.00	1.00	1.00	1.00	1.00	-
	- Yes	1.22 (0.90–1.66)	1.21 (0.89–1.65)	1.16 (0.85–1.59)	0.89 (0.78–1.03)	0.95 (0.81–1.10)	1.01 (0.87–1.18)	0.427
Financial situation, level	- Upper	1.00	1.00	1.00	1.00	1.00	1.00	-
	- Middle	0.94 (0.58–1.52)	0.93 (0.57–1.50)	0.91 (0.56–1.48)	1.48 (1.33–1.65)	1.26 (1.12–1.42)	1.18 (1.04–1.33)	-
	- Lower	0.84 (0.49–1.44)	0.85 (0.49–1.45)	0.88 (0.51–1.53)	1.26 (0.99–1.61)	0.94 (0.72–1.24)	0.84 (0.63–1.11)	0.211
Smoking status	- Never	1.00		1.00	1.00		1.00	-
	- Ex-smoker	1.60 (1.19–2.16)		1.54 (1.14–2.07)	1.95 (1.74–2.18)		1.75 (1.56–1.97)	-
	- Current smoker	0.94 (0.72–1.24)		0.89 (0.67–1.18)	1.87 (1.58–2.21)		1.46 (1.22–1.74)	0.012
Drinking alcohol 2+ times per week	- No	1.00		1.00	1.00		1.00	-
	- Yes	1.30 (0.97–1.74)		1.25 (0.93–1.68)	0.73 (0.66–0.82)		0.81 (0.73–0.91)	0.007
Drinking 5+ alcohol drinks per occasion	- No	1.00		1.00	1.00		1.00	-
	- Yes	1.34 (1.05–1.71)		1.35 (1.06–1.73)	2.35 (2.00–2.78)		1.96 (1.66–2.33)	0.015
(b)								
		**KYH, OR (95% CI)**			**Tromsø7, OR (95% CI)**			***p*-Value for Interaction ^d^**
		**Model 1 ^a^**	**Model 2 ^b^**	**Model 3 ^c^**	**Model 1 ^a^**	**Model 2 ^b^**	**Model 3 ^c^**	
Age, years	- 40–44	1.00	1.00	1.00	1.00	1.00	1.00	-
	- 45–49	1.02 (0.73–1.43)	1.05 (0.75–1.48)	1.06 (0.75–1.49)	1.30 (1.04–1.61)	1.25 (1.00–1.55)	1.26 (1.01–1.56)	-
	- 50–54	1.90 (1.39–2.61)	1.91 (1.39–2.62)	1.98 (1.43–2.73)	1.36 (1.10–1.69)	1.25 (1.01–1.56)	1.26 (1.02–1.57)	-
	- 55–59	3.31 (2.43–4.51)	3.19 (2.33–4.35)	3.56 (2.59–4.90)	2.11 (1.71–2.59)	1.85 (1.50–2.29)	1.89 (1.53–2.34)	<0.001
	- 60–64	3.94 (2.89–5.36)	3.69 (2.71–5.04)	4.38 (3.17–6.04)	2.29 (1.86–2.83)	1.94 (1.56–2.40)	2.01 (1.61–2.49)	-
	- 65–69	5.38 (3.96–7.31)	5.08 (3.73–6.93)	6.10 (4.42–8.43)	3.09 (2.50–3.81)	2.36 (1.90–2.92)	2.49 (1.99–3.10)	-
	*p*-value for trend	<0.001	<0.001	<0.001	<0.001	<0.001	<0.001	-
Education	- College/university	1.00	1.00	1.00	1.00	1.00	1.00	-
	- Primary/secondary	1.48 (1.20–1.82)	1.43 (1.16–1.76)	1.34 (1.08–1.65)	1.66 (1.47–1.86)	1.45 (1.28–1.65)	1.33 (1.17–1.52)	0.981
Living with spouse/partner	- No	1.00	1.00	1.00	1.00	1.00	1.00	-
	- Yes	0.95 (0.80–1.13)	1.02 (0.85–1.21)	1.06 (0.88–1.26)	0.87 (0.76–0.99)	1.01 (0.87–1.16)	1.03 (0.89–1.19)	0.804
Financial situation, level	- Upper	1.00	1.00	1.00	1.00	1.00	1.00	-
	- Middle	1.37 (0.89–2.10)	1.36 (0.89–2.09)	1.30 (0.85–2.01)	1.73 (1.48–2.01)	1.53 (1.30–1.81)	1.44 (1.22–1.70)	-
	- Lower	1.98 (1.26–3.11)	1.92 (1.21–3.03)	1.80 (1.13–2.86)	2.32 (1.87–2.88)	1.89 (1.46–2.43)	1.70 (1.32–2.20)	0.534
Smoking status	- Never	1.00		1.00	1.00		1.00	-
	- Ex-smoker	1.17 (0.90–1.53)		1.11 (0.85–1.46)	1.34 (1.18–1.53)		1.28 (1.12–1.46)	-
	- Current smoker	2.04 (1.59–2.63)		1.79 (1.38–2.33)	1.64 (1.39–1.94)		1.35 (1.13–1.61)	0.077
Drinking alcohol 2+ times per week	- No	1.00		1.00	1.00		1.00	-
	- Yes	1.16 (0.69–1.94)		1.15 (0.68–1.94)	0.65 (0.56–0.74)		0.72 (0.63–0.83)	0.090
Drinking 5+ alcohol drinks per occasion	- No	1.00		1.00	1.00		1.00	-
	- Yes	1.80 (1.31–2.47)		1.48 (1.06–2.05)	1.62 (1.23–2.12)		1.28 (0.97–1.70)	0.521

^a^ Model 1—age-adjusted for all variables except age; ^b^ model 2—adjusted for age and socio-economic variables; ^c^ model 3—adjusted for age, socio-economic variables and health behaviors; ^d^ likelihood ratio test for interaction with “study” variable in model 3 repeated with pooled KYH and Tromsø7 data and the introduced “study” variable.

**Table 4 ijerph-19-09428-t004:** Odds ratios of general and abdominal obesity in Know Your Heart study versus the seventh Tromsø Study with stepwise adjustments for socio-demographic characteristics, lifestyle characteristics and their interactions with the study-defining variable.

	Model 1 ^a^	Model 2 ^b^	Model 3 ^c^
General obesity			
Men	1.09 (0.97–1.23)	1.13 (1.00–1.29)	1.06 (0.92–1.21)
Women	2.20 (2.00–2.43)	2.23 (2.01–2.49)	2.02 (1.80–2.26)
Abdominal obesity			
Men	1.11 (0.98–1.26)	1.15 (1.00–1.32)	0.96 (0.83–1.11)
Women	3.86 (3.49–4.27)	3.88 (3.47–4.34)	3.76 (3.33–4.24)

^a^ Model 1—adjusted for age; ^b^ model 2—adjusted for age and socio-economic covariates; ^c^ model 3—adjusted for age, socio-economic and health behaviors.

## Data Availability

Researchers may apply for access to KYH and Tromsø7 data. See data access regulations and instructions at https://metadata.knowyourheart.science (accessed on 15 May 2022) and https://uit.no/research/tromsostudy (accessed on 15 May 2022), respectively. All data requests will be guided by the protection of personal information, the confidentiality agreement with the participants and participants’ informed consent.

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
