# Peer review of "Obesity Prevalence and Associated Socio-Demographic Characteristics and Health Behaviors in Russia and Norway"

_ijerph, 2022, doi:10.3390/ijerph19159428_

Round 1

Reviewer 1 Report

This is an interesting study to investigate the association of obesity (GO, AO) with socio-demographic factors in two cohorts.

Information relevant to variables was provided with clarity and a reasonable categorization definition.

The associations of these variables with GO and AO were also examined using unadjusted and adjusted models and reported with necessary justification. Although most of the variables did not display a strong association with GO or AO, some interesting results were also reported.

This reviewer also appreciates the thorough discussion on the potential limitations of this study, which may contribute to moderate associations with sex and country differences.  

Since the main variable in this study, financial situations, is categorized into 7 levels that primarily rely on self-reported reflection, it may also contribute to the weak association with obesity.

Overall, the study provided a thorough comparison with a relatively small list of variables; however, the choice/availability of data from the two cohorts only yielded new information in the context of obesity and socio-demographics.

Reviewer 2 Report

Dear authors,

Here are some recommendations for change.

Best regards.

Methodology section:

Section 2.2 indicates the measurement protocol for several anthropometric measurements. However, it does not describe the weight and height measurement protocol, if the authors want the reviewers to review articles 33, 37 or 38 they must indicate on which page the information of interest is located. I recommend that you review the journal's citation rules.

In section 2.4, which deals with health behaviors, the authors must justify why they used the variables described and not others, or if this information is found in the original references, the page number where it can be reviewed must be indicated. In the event that this justification has not previously been made, it should be put as a limitation of this article in the discussion section. Two examples are indicated where the components to observe the health of a subject are justified at a theoretical level:

a. On page e13 of this reference, the components of health are indicated to identify when a subject has ideal cardiovascular health or not (Roger VL, Go AS, Lloyd-Jones DM, et al. Heart disease and stroke statistics--2012 update : a report from the American Heart Association. Circulation. 2012;125(1):e2–e220).

b. This article describes the necessary components to recognize whether or not the Mediterranean lifestyle is followed (Sotos-Prieto M, Moreno-Franco B, Ordovás JM, León M, Casasnovas JA, Peñalvo JL. Design and development of an instrument to measure overall lifestyle habits for epidemiological research: the Mediterranean Lifestyle (MEDLIFE) index. Public Health Nutr. 2015 Apr;18(6):959-67)

Results section:

The tables must be self-explanatory, so it must be indicated if the results are expressed by mean and standard deviation. It would be advisable to change the value of Abs to n (number of the sample).

Bibliography section:

99 references were found.

It has approximately 25 citations that are more than 10 years old.

Review the Vancouver standards of the magazine in the references used, errors have been identified.
